# Hereditary Hemorrhagic Telangiectasia (HHT) and Survival: The Importance of Systematic Screening and Treatment in HHT Centers of Excellence

**DOI:** 10.3390/jcm9113581

**Published:** 2020-11-06

**Authors:** Els M. de Gussem, Steven Kroon, Anna E. Hosman, Johannes C. Kelder, Martijn C. Post, Repke J. Snijder, Johannes J. Mager

**Affiliations:** 1Department of Internal Medicine, Division of Respirology, University of Manitoba, Grace Hospital, 400 Booth Drive, Winnipeg, MB R3J 3M7, Canada; edegussem@hsc.mb.ca; 2Department of Pulmonology, St Antonius Hospital, Koekoekslaan 1, 3435 CM Nieuwegein, The Netherlands; s.kroon@antoniusziekenhuis.nl (S.K.); a.hosman@antoniusziekenhuis.nl (A.E.H.); r.snijder@antoniusziekenhuis.nl (R.J.S.); 3Department of Epidemiology and Medical Statistics, St Antonius Hospital, Koekoekslaan 1, 3435 CM Nieuwegein, The Netherlands; keld01@antoniusziekenhuis.nl; 4Department of Cardiology, St Antonius Hospital, Koekoekslaan 1, 3435 CM Nieuwegein, The Netherlands; m.post@antoniusziekenhuis.nl; 5Department of Cardiology, Utrecht University Medical Center, Heidelberglaan 100, 3584 CX Utrecht, The Netherlands

**Keywords:** telangiectasia, hereditary hemorrhagic, vascular malformations, survival, life expectancy

## Abstract

Hereditary hemorrhagic telangiectasia (HHT), an autosomal dominant disease, is characterized by telangiectases and arteriovenous malformations (AVMs). Untreated AVMs, especially in the lungs—pulmonary AVMs (PAVMs)—can result in morbidity with a decreased life expectancy. We have investigated whether HHT patients, systematically screened for HHT-related organ involvement and treated if needed, have a similar survival as persons without HHT. We included all individuals screened for HHT between 2004 and 2016 with a genetically or clinically confirmed diagnosis (HHT group) or excluded diagnosis (non-HHT control group). The social security number was used to confirm status as dead or alive in December 2019. We included 717 HHT patients and 471 controls. There was no difference in survival between the HHT and the non-HHT control group. The HHT group had a life expectancy of 75.9 years (95% confidence interval (CI) 73.3–78.6), comparable to the control group (79.3 years, 95% CI 74.8–84.0, Mantel–Cox test: *p* = 0.29). In conclusion, the life expectancy of HHT patients systematically screened for HHT-related organ involvement and treated if needed in an HHT center of excellence was similar compared to their controls, justifying systematic screening and treatment in HHT patients.

## 1. Introduction

Hereditary Hemorrhagic Telangiectasia (HHT) is an autosomal dominant disease, with prevalence rates between 1:5000 and 1:8000 and with approximately 85,000 affected citizens in Europe [1,2]. HHT is characterized by multi-systemic vascular lesions, known as telangiectases, and visceral arteriovenous malformations (AVMs). In approximately 85% of the HHT patients, mutations in the Endoglin (*ENG*) or Activin receptor-like kinase 1 (*ACVRL1*) gene are found, causing HHT type 1 and type 2, respectively [3,4]. Most of the HHT patients suffer from recurrent, spontaneous epistaxis due to rupture of the thin-walled nasal telangiectases. The visceral AVMs are usually asymptomatic but can result in severe morbidity and mortality. The most common visceral localization of the AVMs is the lung (pulmonary AVM (PAVM)). Although much rarer than PAVMs, cerebral vascular malformations (CVMs) can also result in severe morbidity and mortality. The prevalence of PAVMs and CVMs depends on the HHT type: PAVMs occur in up to 60% of patients with HHT type 1 and 5–10% of patients with HHT type 2, and CVMs occur in 8–16% in HHT type 1 and in 0.5–1.5% in HHT type 2 [5]. A PAVM is a direct connection between the pulmonary artery and pulmonary vein with the absence of a normal capillary bed. Due to the absence of the normal capillaries, septic or non-septic emboli can enter the systemic circulation resulting in strokes and brain abscesses [6]. Preventatively, PAVMs can be safely and effectively treated with transcatheter embolotherapy. For patients with CVMs, treatment versus conservative management, should be considered on a case-by-case basis [7].

In the past, HHT patients presented with major complications especially from undiagnosed PAVMs such as hemothorax and paradoxical emboli leading to ischemic stroke or cerebral abscess [6,8]. These complications significantly reduced the quality of life and the life expectancy [9,10,11]. Several studies have shown a decreased life expectancy in HHT patients that did not receive a systematic HHT screening and treatment [10,12]. According to the current International HHT Guidelines screening and, if indicated, treatment of HHT-related organ involvement is strongly recommended in a center with HHT expertise [7]. The intention of this study is to evaluate if by using this approach the life expectancy of HHT patients is no longer negatively affected.

## 2. Materials and Methods

### 2.1. Study Design and Patient Selection

This retrospective cohort study included all consecutive persons who were referred to our HHT outpatient clinic suspected of having HHT between January 2004 and November 2016. This period was chosen because prior to 2004, HHT screening was not standardized in our center. We used the Dutch social security numbers (SSN) of each patient to confirm status alive or deceased, and date of death with the Dutch Ministry of Healthcare. This check was performed on 1 December 2019.

### 2.2. Patient Selection

We used our HHT database as saved on 14 November 2016 to select the individuals to be included in this study. Patients with a genetic diagnosis (a disease-causing mutation in *ENG*, *ACVRL1* or *SMAD4* gene) and/or clinically confirmed HHT diagnosis according to the Curaçao criteria [3,4], [13,14] were included in the HHT group. Patients without a clinical diagnosis (e.g., none or only 1 criterion) and patients with a possible diagnosis (2 criteria) but without a disease-causing mutation in *ENG*, *ACVLR1* or *SMAD4* (or in the absence of the known HHT family mutation) were included in the non-HHT control group. Patients with two positive clinical criteria but without DNA testing or with a family member with an unknown type of HHT were excluded. The Curaçao criteria include (1) the presence of spontaneous, recurrent epistaxis; (2) multiple mucocutaneous telangiectases at characteristic sites; (3) the presence of visceral AVMs and (4) a first-degree relative with definite HHT [13]. The clinical HHT diagnosis is “unlikely” with less than 2 criteria present, “possible or suspected” with 2 criteria present and “definite” with three or more positive criteria. DNA testing was (usually) only offered to adult patients. We excluded all patients without a known SSN. Patients with a clinical HHT diagnosis (e.g., three or more positive criteria), but negative genetic testing for the known family HHT mutation were excluded because of the uncertainty of their HHT status. To avoid asymptomatic HHT patients being included in the non-HHT control group, any person who had been screened during childhood, but had not been rescreened as an adult or had not undergone DNA testing as a child, was excluded because of age dependent symptom penetrance and the consequently lower sensitivity of the Curaçao criteria in children [15]. Finally, we excluded patients with insufficient data to confirm or reject the HHT diagnosis or patients who did not complete the screening program. We did not exclude patients that were lost to follow-up after completing the initial screening. 

### 2.3. Screening Protocol for HHT

All patients were screened according to our standardized protocol. This protocol for adults entailed a detailed patient interview, a physical examination focused on signs and symptoms of HHT and PAVMs, an inspection of the nasal mucosa by a dedicated HHT Ear, Nose and Throat specialist, laboratory testing for anemia and a transthoracic contrast echocardiography (TTCE) to screen for the presence of a right-to-left shunt secondary to PAVMs. In case of moderate or severe shunt grade on TTCE [16], a non-contrast chest CT-scan was performed. In case of a clinically or genetic confirmed HHT diagnosis, adult patients were advised to undergo further evaluation for CVMs with a non-contrast magnetic resonance imaging of the brain. Screening for other visceral organ involvement was only done on indication. For example, the digestive tract was evaluated in cases with anemia not correlated to epistaxis severity, and screening for hepatic vascular malformations (HVMs) was done in cases with elevated liver enzymes, dyspnea or signs and symptoms associated with liver disease or high-output heart failure. Other aspects of HHT such as epistaxis and anemia were treated accordingly. Children were screened with a different protocol that included a detailed history to detect epistaxis or hypoxemia-related symptoms such as exercise intolerance, poor growth or headaches, a chest radiography and pulse-oximetry. Children were only screened for CVMs on indication. Further investigation with a low-dose chest CT-scan was only performed when abnormalities were found suspect for the presence of a PAVM: a suspect history for PAVMs, saturation with pulse oximetry <96% or a density suspect for a PAVM on the chest radiography [17]. If a treatable PAVM was detected based on a discussion in the multidisciplinary team, embolotherapy of the PAVM with vascular plugs or coils was performed by interventional radiologists specialized in HHT. PAVMs with a feeding artery ≥2–3 mm were regarded as treatable. After embolization of all targeted PAVMs, patients were reviewed in the multidisciplinary team. Follow-up in our institution is standardized and includes a contrast-enhanced chest CT-scan 6 months after embolization, followed by a chest CT-scan every 2 to 5 years in case of sustained occlusion of the embolized PAVM. The follow-up chest CT-scans were discussed in the multidisciplinary team. In case of persistent perfusion or reperfusion of the PAVM on follow-up chest CT-scan, patients were scheduled for repeat embolotherapy. Signs that suggest persistent perfusion or reperfusion include contrast enhancement in the PAVM, no or minimal shrinkage of the PAVM sac or persistence of a large feeding artery or draining vein. Children were followed on a case-by-case basis after embolotherapy, usually with a chest radiography and saturation measurement with pulse oximetry. If a CVM was detected, the patient was subsequently referred to a center with expertise in treating brain vascular malformations in The Netherlands. The need for treatment and follow-up was determined in this center. If no abnormalities were detected with screening in children or adults, rescreening was performed every five years. All children were advised to undergo rescreening with our protocol for adults when they reached the age of 18 years.

### 2.4. Statistical Methods and Ethics

Statistical analysis was performed using SPSS version 26.0 for Windows (IBM, Armonk, NY, USA) and R version 3.5.3 for Windows (the R Project for Statistical Computing). Data are presented as mean and standard deviation (SD). Continuous variables were compared using the independent samples T-test. The prevalence of anemia, comorbidities and disease complications were compared using Fisher’s exact test. Survival was estimated using left-truncated Kaplan–Meier curves. For comparison of survival between groups we used the Mantel–Cox test. Statistical significance was defined at *p* < 0.05. This study was approved by the research ethics board of Medical Research Ethics Committees United (MEC-U) of the St. Antonius Hospital under protocol registration number W16.160.

## 3. Results

### 3.1. Patient Selection and Baseline Characteristics

In total, 1541 individuals had been screened for presence of HHT between 2004 and November 2016. From these individuals, 717 patients could be included in the HHT group and 471 in the non-HHT control group. In the HHT group, 319 patients (45%) suffered from HHT type 1, 325 (45%) from HHT type 2, 29 (4%) from juvenile polyposis/HHT overlap syndrome and in 44 patients no disease-causing mutations could be identified. The mean age at presentation in the HHT group was 40.6 years and 54% was female. In the control group the mean age was 40.9 years with 57% female. The mean birthyear for both groups was the year 1969. The control group consisted of family members of HHT patients in whom HHT was ruled out (*n* = 368), patients with recurrent epistaxis (*n* = 28) without HHT, patients with mucocutaneous telangiectases without HHT (*n* = 17) or patients with the suspicion of visceral AVM (*n* = 58) without HHT. The suspected visceral AVMs (not all were confirmed) were located in the lungs (PAVM: *n* = 43), digestive tract telangiectases (*n* = 5), brain (CAVM: *n* = 4) and four suspected AVMs were located in other organs. In Figure 1 the flowchart of the patient selection is depicted. The demographic characteristics of the included subjects are shown in Table 1.

### 3.2. Visceral AVMs, Comorbidities and Disease Complications

In Table 2, the number of visceral AVMs, comorbidities and disease complications of the HHT patients and controls are shown. In the HHT group, 255 patients (36%) had PAVMs versus 32 (7%) individuals in the control group. In the HHT group, 60% (432 patients) underwent CVM screening. In 28 out of 432 patients (6%) a CVM was detected. Furthermore, (symptomatic) HVMs and gastrointestinal telangiectases were diagnosed in 75 (11%) and 72 (10%) patients, respectively. The prevalence of anemia was significantly higher in the HHT group compared to the controls (*p* < 0.001). Comorbidities and disease complications were similar between groups with the exception of higher prevalence of pulmonary hypertension (3% vs. <1%, Fisher’s exact test: *p* = 0.003) and high-output heart failure (Fisher’s exact test: 2% vs. 0%, *p* = 0.001) in the HHT group. 

### 3.3. Survival of HHT Patients and Controls

In Table 3, the number of patients, and the age and cause of death are shown. In the HHT group 57 patients (8%) had died versus 24 (5%) in the control group. The most frequent cause of death in the HHT was infectious disease, followed by malignancy. The cause of death in the control group was not recorded. There was no difference in survival between the HHT and the non-HHT control group (Mantel–Cox test: *p* = 0.29; Figure 2). The survival of patients with HHT type 1 did not differ from patients with HHT type 2 (Mantel–Cox test: *p* = 0.28; Figure 3A). Compared to the non-HHT control group, the survival of both patients with HHT type 1 (Mantel-Cox test: *p* = 0.28) and patients with HHT type 2 (Mantel–Cox test: *p* = 0.85) did not differ (Figure 3B,C). The mean life expectancy of the HHT population was 75.9 years (95% CI 73.3–78.6 years), comparable to the non-HHT control group (79.3 years, 95% CI 74.8–84.0 years). The mean life expectancy for patients with HHT type 1 was 76.4 years (95% CI 71.6–82.3 years) and 77.9 years (95% CI 74.5–81.3 years) for patients with HHT type 2. The survival of patients with a genetically confirmed HHT diagnosis (HHT type 1, type 2 or juvenile polyposis/HHT overlap syndrome) was comparable to the survival of their relatives in whom HHT was ruled out (e.g., a known family mutation, the individual did not inhere the specific mutation) (Mantel–Cox test: *p* = 0.43; Figure 3D). There was no significant difference in survival of men and women with HHT (Mantel–Cox test: *p* = 0.10; Figure 4A). HHT patients with visceral AVMs had a worse survival compared to HHT patients without visceral AVMs (Mantel–Cox test: *p* = 0.017). The survival in HHT patients with and without PAVM, CVM or gastrointestinal telangiectases did not differ, only patients with HVM had significantly worse survival compared to HHT patients without HVM (see Figure 4B–F). HHT patients without anemia showed a tendency to a better survival (see Figure 4G).

## 4. Discussion

We found that the life expectancy of patients with HHT who have been systematically screened for HHT-related organ involvement, treated if needed and followed in a center with HHT expertise did not differ from the life expectancy of the non-HHT control group. These results emphasize the importance of systematic screening of HHT patients, treatment and follow-up in an HHT expertise center.

Previous studies have shown conflicting results on life expectancy in patients with HHT. A study with 73 patients with HHT and 218 controls with a 20-year follow-up did not reveal any significant differences in survival between both groups [18]. However, two studies assessing survival in parents of a current HHT population showed worse survival in these parents with HHT compared to their non-HHT partners [10,12]. Since the population of the latter studies was comprised of parents with HHT, these studies concerned a largely unscreened and untreated population. In the study by De Gussem et al., patients with HHT type 1 had especially worse survival compared to non-HHT partners. The authors stated that this was probably caused by the higher prevalence of (untreated) PAVMs and CVMs and subsequent complications [10]. Donaldson et al. observed higher risks of stroke, cerebral abscess and bleeding complications and showed poorer survival in a study on a primary care database including almost 700 HHT patients in comparison to age and sex matched controls [9]. However, as emphasized by the authors, many of the HHT-related complications were amenable to intervention, early diagnosis and treatment. Their study also showed a higher mortality rate in the time period closest to the diagnosis of HHT. This might be related to the fact that some patients presented with HHT-related complications at the time of diagnosis. Once patients have been diagnosed with HHT, they are followed and treated for organ involvement to reduce morbidity and mortality. 

Our study shows that the life expectancy of patients, who are screened for HHT-related organ involvement and if needed, treated in a center with HHT expertise, is similar to the life expectancy of their relatives without HHT. We did not observe difference in survival between HHT type 1 and type 2. This is an important finding because a previous study by our group showed worse survival in (largely untreated) HHT type 1 population [10]. In addition, survival of men and women with HHT and HHT patients with and without PAVMs, CVMs or gastrointestinal telangiectases did not differ. We observed a significant difference in HHT patients with and without visceral AVMs and patients with and without HVMs. This is probably caused by the fact that screening for HVMs is only done on indication in our center and thus the majority of these patients probably suffered from symptomatic HVMs.

This improvement in survival of HHT patients in comparison to previous publications is most likely due to the benefit of screening for the presence of PAVMs and PAVM treatment if indicated. Additionally, screening methods and treatment options for PAVMs, CVMs, epistaxis, gastro-intestinal bleeding and subsequent anemia have improved over the years, which most likely benefits survival as well. In addition to this, several studies have suggested that patients with HHT possibly have a natural protection against certain cancers and myocardial infarction, possibly benefiting survival rates [19,20]. While we did not observe a negative effect of HHT on the survival in this study, it is important to realize that the quality of life in these patients will probably still be lower compared to their controls even if optimal screening and treatment was performed. Two studies investigated the quality of life in an HHT population screened and treated in two different HHT centers. The authors observed that the HHT-related symptoms and organ involvement had a major negative effect on the quality of life [21,22]. 

We have not compared the survival of the HHT patients and controls to the Dutch population. However, we think that they are in line with each other. The life expectancy of Dutch people born in 1969 was 73.54 years [23]. We observed a slightly higher life expectancy of our HHT patients and controls, most likely because the patients included in this study already lived to an average age of approximately 40 years. The comorbidities were comparable between the HHT group and the controls with the exception of anemia, higher rates of pulmonary hypertension and high-output heart failure in the HHT group. It would be very informative to compare our HHT group, to HHT patients that did not receive screening; however, this is not ethically possible.

We acknowledge that there is selection bias of the patients who passed away prior to the HHT diagnosis and were thus not included in this study. Additionally, there is a small chance of presence of undiagnosed HHT patients in the non-HHT control group. However, in view of the number of patients and non-HHT controls in this study and in view of the convincing results, it seems unlikely that this would influence the outcomes significantly. Strengths of this study are the use of consecutive data in a large patient population. Although the control group also consists of patients that were referred to our hospital with visceral AVMs, the largest part of the controls included first-degree family members of HHT patients. Therefore, the lifestyle and socio-economic conditions are to some degree comparable which makes the non-HHT controls a better control group than the Dutch population for evaluation the influence of HHT on life expectancy. 

In conclusion, the survival of patients with HHT is not negatively affected by HHT if the HHT patients have been systematically screened and treated for HHT-related organ involvement in a center with HHT expertise. These findings demonstrate the importance of systematic screening of HHT patients and treatment of PAVMs and other HHT-related organ involvement.

## Figures and Tables

**Figure 1 jcm-09-03581-f001:**
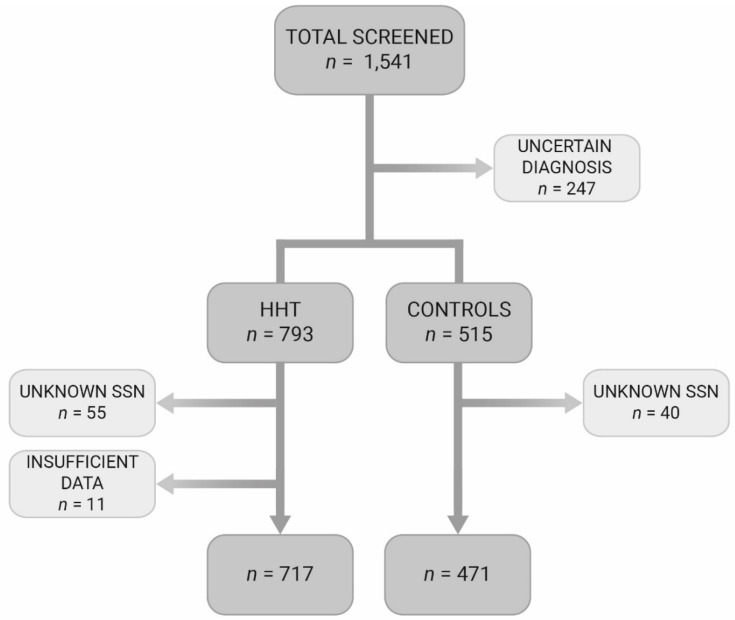
Flowchart of included patients. HHT, Hereditary Hemorrhagic Telangiectasia; SSN, social security number.

**Figure 2 jcm-09-03581-f002:**
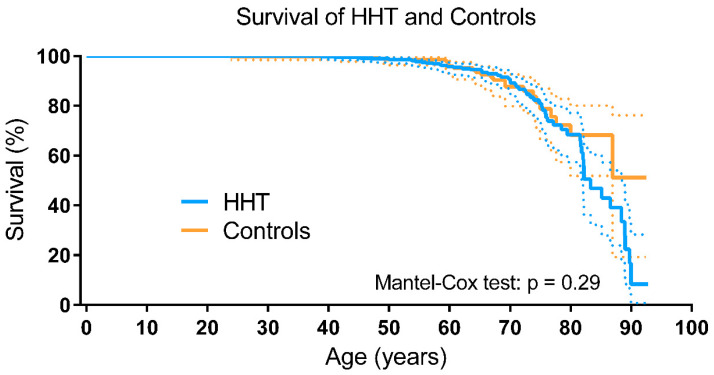
Left-truncated Kaplan–Meier curve of HHT and controls. The dotted lines represent the 95% confidence intervals. HHT, Hereditary Hemorrhagic Telangiectasia.

**Figure 3 jcm-09-03581-f003:**
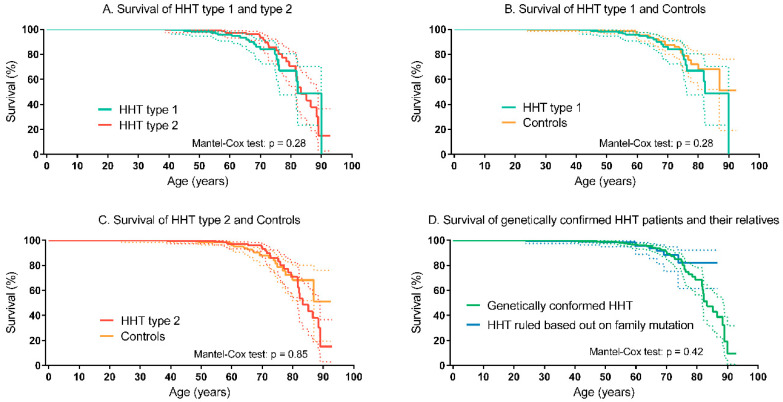
Left-truncated Kaplan–Meier curves of HHT subtypes and controls (**A**) HHT type 1 and HHT type 2. (**B**) HHT type 1 and controls. (**C**) HHT type 2 and controls. (**D**) Patients with genetically confirmed HHT (HHT type 1, type 2 and juvenile polyposis/HHT overlap syndrome) and their relatives without HHT. The dotted lines represent the 95% confidence intervals. HHT, Hereditary Hemorrhagic Telangiectasia.

**Figure 4 jcm-09-03581-f004:**
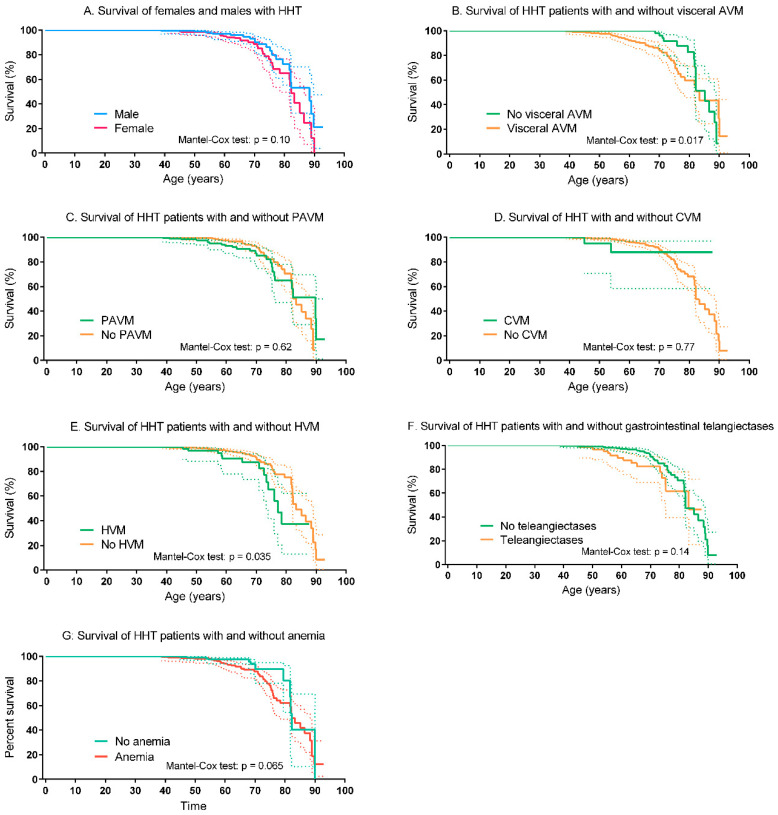
Left-truncated Kaplan–Meier curves of HHT patients. (**A**) Survival of females and males with HHT. (**B**) Survival of HHT patients with and without visceral AVM. (**C**) Survival of HHT patients with and without PAVM. (**D**) Survival of HHT patients with and without CVM. (**E**) Survival of HHT patients with and without HVM. (**F**) Survival of HHT patients with and without gastrointestinal telangiectases. (**G**) Survival of HHT patients with and without anemia. The dotted lines represent the 95% confidence intervals. AVM, arteriovenous malformation; CVM, cerebrovascular malformation; HHT, Hereditary Hemorrhagic Telangiectasia; HVM, hepatic vascular malformation.

**Table 1 jcm-09-03581-t001:** Demographic characteristics of the included patients. *ACVRL1*, activin receptor-like kinase 1 (HHT type 2); *ENG*, endoglin (HHT type 1); HHT, Hereditary Hemorrhagic Telangiectasia; SD, standard deviation; *SMAD4*, SMAD family member 4 (juvenile polyposis/HHT overlap syndrome).

	HHT Group (*n* = 717)	Control Group (*n* = 471)	*p*-Value
Gender (%)			0.28
Female	384 (54)	268 (57)	
Male	333 (46)	203 (43)	
Genetic mutation (%)		-	-
*ENG* (HHT type 1)	319 (45)		
*ACVRL1* (HHT type 2)	325 (45)		
*SMAD4*	29 (4)		
Mutation unknown	44 (6)		
Mean age at presentation, years (SD)	40.8 (19.4)	40.6 (17.4)	0.86
*ENG* (*n* = 319)	35.7 (19.9)	-	-
*ACVRL1* (*n* = 325)	44.9 (17.8)	-	-
*SMAD4* (*n* = 29)	31.4 (17.2)	-	-
Mutation unknown (*n* = 44)	54.1 (13.9)	-	-
Mean birth year	1969	1969	-

**Table 2 jcm-09-03581-t002:** Visceral AVMs and comorbidities of the included patients. *ACVRL1*, activin receptor-like kinase 1 (HHT type 2); AVM, arteriovenous malformation; COPD, chronic obstructive pulmonary disease; CVM, cerebrovascular malformations; *ENG*, endoglin (HHT type 1); HHT, Hereditary Hemorrhagic Telangiectasia; HVM, hepatic vascular malformation; SD, standard deviation; *SMAD4*, SMAD family member 4 (juvenile polyposis/HHT overlap syndrome). * For these AVMs, screening is only performed on indication.

	HHT Group (*n* = 717)	Control Group (*n* = 471)
PAVM (%)	255 (36)	32 (7)
*ENG* (*n* = 319)	176	-
*ACVRL1* (*n* = 325)	47	-
*SMAD4* (*n* = 29)	12	-
Mutation unknown (*n* = 44)	20	-
PAVM embolotherapy	175	28
CVM (%)		
Yes	28 (4)	4 (2)
No	404 (56)	0
Unknown/not screened	285 (40)	467 (99)
CVM treatment		
No treatment	14	1
Surgery	5	1
Radiotherapy	3	1
Embolotherapy	2	1
Combination	4	0
HVM (%) *	75 (11)	3 (<1)
Gastrointestinal telangiectases (%) *	72 (10)	6 (1)
Other AVMs (%) *	14 (2)	4 (1)
Spinal	4	0
Pancreatic	3	0
Renal	2	1
Urinary bladder	2	1
Splenic	1	0
Muscular	1	2
Ocular	1	0
Anemia (%)	212 (30)	28 (6)
Comorbidities (%)		
Malignancy	36 (5)	21 (5)
Atrial fibrillation	36 (5)	13 (3)
COPD and bronchiectasis	28 (4)	16 (3)
Acute coronary disease	26 (4)	10 (2)
Venous thromboembolism	25 (4)	8 (2)
Autoimmune disease	21 (3)	23 (5)
Pulmonary hypertension	19 (3)	2 (< 1)
Diabetes mellitus type 2	16 (2)	19 (4)
Peripheral vascular disease	15 (2)	7 (2)
Disease complications (%)		
Cerebrovascular accident	49 (7)	22 (5)
High-output heart failure	14 (2)	0 (0)
Brain abscess	11 (2)	3 (1)

**Table 3 jcm-09-03581-t003:** Number of deceased patients, age and cause of death. HHT, Hereditary Hemorrhagic Telangiectasia; SD, standard deviation.

	HHT Group (*n* = 717)	Control Group (*n* = 471)	*p*-Value
Deceased (%)	57 (8)	24 (5)	0.06
Mean age death, years (SD)	69.7 (13.3)	64.3 (13.7)	0.1
Cause of death		-	-
Infection	15		
Heart failure	9		
Malignancy	8		
Severe anemia	4		
Thromboembolism	2		
Postoperative complications	2		
Hemorrhagic stroke	1		
Unknown	1		
	17		

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
