# Peer review of "Hereditary Hemorrhagic Telangiectasia (HHT) and Survival: The Importance of Systematic Screening and Treatment in HHT Centers of Excellence"

_jcm, 2020, doi:10.3390/jcm9113581_

Round 1

Reviewer 1 Report

Although very interesting some important issues have to be addressed:

  1. The control group is not really a representation of the healthy Dutch population. There are several biases by taking this heterogenic group - that at least part were referred to this specific clinic because of a pathology- e.g.-in the control group there are 7%!! With PAVMs- we can not regard them as normal controls?  this is not the prevalence of PAVMs in the healthy population- There is definitely a referral bias. Maybe some of them died earlier because of this. Ideally you should have taken only the healthy family members- but if you take the whole group it should be discussed as a limitation. 
  2. In general I think that a paragraph explaining the limitations should be at the end of the manuscript.
  3. Table 1- needs p values for the relevant parameters
  4. Table 3- same
  5. The average age of death in the study was 64-69. The average age of death in the Netherlands is 78years- Am I mistaken? https://www.statista.com/statistics/521093/netherlands-average-age-at-death-by-gender/

Author Response

Reviewer #1

Although very interesting some important issues have to be addressed:

  1. The control group is not really a representation of the healthy Dutch population. There are several biases by taking this heterogenic group - that at least part were referred to this specific clinic because of a pathology- e.g.-in the control group there are 7%!! With PAVMs- we can not regard them as normal controls? this is not the prevalence of PAVMs in the healthy population- There is definitely a referral bias. Maybe some of them died earlier because of this. Ideally you should have taken only the healthy family members- but if you take the whole group it should be discussed as a limitation.

This is a good point raised by the reviewer. We have performed a sub analysis and compared the genetically confirmed HHT patients with their healthy family members (in which HHT has been ruled out based on the known family mutation). There was no difference in survival (p = 0.42) between both groups.

We have added the following to the result section (p. 11, line 15-18):

The survival of patients with a genetically confirmed HHT diagnosis (HHT type 1, type 2 or juvenile polyposis/HHT overlap syndrome) was comparable to the survival of their relatives in whom HHT was ruled out (e.g., a known family mutation, the individual did not inhere the specific mutation) (Mantel-Cox test: p = 0.43; figure 3D). 

We have added an additional graph to figure 3 (figure 3D).

We have added the following to the discussion (p. 15, line 8-10):

Although the control group also consists of patients that were referred to our hospital with visceral AVMs, Tthe largest part of the controls included first-degree family members of HHT patients.

  1. In general I think that a paragraph explaining the limitations should be at the end of the manuscript.

The paragraph with limitations has been added to the end of the manuscript

  1. Table 1- needs p values for the relevant parameters

We have added the P-values for the relevant parameters to table 1.

  1. Table 3- same

We have added the P-values for the relevant parameters to table 3.

  1. The average age of death in the study was 64-69. The average age of death in the Netherlands is 78years- Am I mistaken? https://www.statista.com/statistics/521093/netherlands-average-age-at-death-by-gender/

We understand the point raised by the reviewer. Indeed, the average age of the patients who died during the study period, is lower than these data. However, it should be noted that in our study only 8% of the HHT patients and 5% of the controls have died: most are still alive and many of them are now older than 65. So, the average age of death in this study can not be regarded as the average age of death of the study population, simply because more than 90% is still alive. We have estimated the life expectancy of all HHT patients and controls. These data, as mentioned in the discussion, are quite comparable to the Dutch population born in 1969.  

Reviewer 2 Report

The Authors clarified to the required issues.
It should be noted:

In table n°2: put the percentage also for the CMV category.

In table n°3: It would be better  to distinguish between comorbidities and disease complications.

Finally, I would not include "cardiac arrest" among the causes of death.

Author Response

Reviewer #2

The Authors clarified to the required issues.

It should be noted:

In table n°2: put the percentage also for the CMV category.

We have added the percentages for CVM category in table 2.

In table n°3: It would be better  to distinguish between comorbidities and disease complications.

We have changed table 3 and we have distinguished between comorbidities and disease complications.

Finally, I would not include "cardiac arrest" among the causes of death.

We have excluded ‘cardiac arrest’ as cause of death. In these two cases, the exact cause of the cardiac arrest was unknown. We have added these patients to the category ‘unknown’.

Reviewer 3 Report

This manuscript was well revised, and it can be accepted.

Author Response

Thank you!

Reviewer 4 Report

The authors have addressed my concerns very well. I have no further comments.

Author Response

Thank you!

This manuscript is a resubmission of an earlier submission. The following is a list of the peer review reports and author responses from that submission.

Round 1

Reviewer 1 Report

The authors' aim to provide information regarding life expectancy in HHT is extremely important. As previous studies suggested that HHT is associated with shorter life expectancy- presenting normal life expectancy in screened patients will motivate patients and caregivers to screen patients. 

Re the manuscript:

Introduction: is comprehensive and clear

Methods: the methods are clear, however several minor points have to clarified: 

  1. The non HHT group
    1. What was the reason for referral- just family history or maybe other reasons - GI bleed? Liver AVM? Brain AVM? which might pose them to other morbidities and shortened life expectancy
    2. What did you do with patients with 2 criteria- were they excluded? Are these the "uncertain diagnosis"
  2. The HHT group
    1. Did all patients get full screening – no dropouts? Lost to follow-up? Refusal? What did you do with these patients?

Results and discussion are clear and the main aim of the study is well presented in the results. However I think several points have to be discussed: 

Life expectancy in the Netherlands is 81.56y in 2017. An argument might be that the data are from 2004 and not only recent years- the life expectancy in 2010 was 80.18y. Although no statistical calculation was made but it seems that the results of the study cohort are not as good. 

Mean age of death was also younger than reported in the Netherlands for the general population. 

I believe these issues have to be discussed.

Minor points: 

Are people with no SSN have certain characteristics (immigrants etc.)- is there an option that they could have drop the life expectancy (in both groups)

In the study there is female predominance. Women have better life expectancy Was there a difference in life expectancy between women and men in HHT? 

Reviewer 2 Report

Overall, good effort in taking advantage of what sounds like a rich and valuable data set that has the potential to make an important contribution in this area of investigation.

Unfortunately, as presented, this manuscript is not publishable without significant editing to create focus, clarity, and consistency.

While the title and much of the manuscript discusses “screening”, the cohort of HHT patients that was analyzed was not only “screened”, but they were also likely “treated” for the abnormalities that were found on screening and subsequent rescreening.  Therefore, any effect of their “intervention” must be attributable to “screening and treatment” unless they have data to suggest otherwise. While the authors talk about some aspects of treatment in the manuscript, their discussion is inconsistent and does not assign enough credit to the possibility that “treatment” in addition to “screening” contributed to the results presented. Unfortunately, it took the authors until the last sentence of the manuscript to state what the manuscript is really all about and what needs to presented in the title, abstract and rest of the manuscript: “the importance of systematic screening of HHT patients and treatment of PAVMs and other HHT related organ involvement.”

Along the same lines of not including potentially important aspects of HHT screening and treatment, the authors do not present information to defend why they mostly limit their discussion of screening and treatment to PAVMs and not on all of the other aspects of HHT that their patients were probably screened and treated for, including but not limited to CVMs, iron deficiency anemia, HAVMs, epistaxis. Unlikely that there is any way to separate out the impact of any one of a number of things that were done, therefore, details of everything that was done to screen and treat must be included, otherwise the naïve reader has potential to think that focused screening and treatment is sufficient.  Again, this report is really about comprehensive HHT screening, treatment and follow-up at a well-established HHT Center of Excellence – this needs to be discussed and clearly stated/reinforced.

Some more specific concerns:

Control group: For a numbers of reason, I like the “control” cohort that the authors use, however I think it is important and valuable to clearly state that no difference in life expectancy between subjects with HHT who have been screened and treated and persons who have been screened and found not to have HHT because they had a risk factor for HHT that resulted in screening for HHT; their “control” group is not a group of people with HHT who have not been screened (which would be a better, but not ethical, control group to make their point) nor is it a group of people from the general population (without a risk factor that resulted in screening for HHT) who have never been screened for HHT.

Title:  need a clear, concise statement of results.  Stating a question in the title is not helpful.

Survival vs Life Expectancy.  I am not an expert in this area, so I may be off base on what follows. However, in the manuscript as presented, I feel that the use of these terms need to be more thoughtfully considered and presented because I am confused by what the authors are trying to say. I think the authors want to make the point that no difference in “life expectancy” between the two cohorts and that this may need to be presented as “mean life expectancy”. With this in mind, legend for Fig 2 needs clarity: what kind of survival plots?

HHT1 vs HHT 2 data presentation and analysis: while potentially important information, the manner in which the data is present and discussed is very superficial and reads as a distracting, poorly developed late addition.  Either delete, or present and discuss in a more meaningful manner.

Large number of subjects in both cohorts with “Unknown SSN”:  seems like a large number of subjects without data.  Also larger % of controls than HHT subjects, therefore ? larger impact on the control data. At a minimum, this needs to be discussed and I suspect that are statistical approaches to analyze the impact of these subjects on the data.

Discussion:  Good start but needs to be clearer, better organized and include additional discussion relevant to above.

Reviewer 3 Report

Useful snd interesting paper that enphatises the role of multidisciplinary approach for The management of these frial and complex patients.

The Authors could provide data about comorbidities in the HHT gruppo vs. control group.

Morover, in The “patients section” I’m not surr I understand  if Inclusion criteria is 3 + Curaçao criteria AND genetic or not.

Reviewer 4 Report

Materials and methods
Screening protocol for HHT
1. Line 107 Was the screening for CVMs non-contrast MRI?
2. Line 113 Please mention about definition of “treatable PAVM?” The size of feeding artery > 3 mm?
3. Line 118 What was definition of persistence in CT diagnosis.
4. How about the strategy of treatment and follow-up for CVMs?

Results
5. I think authors should have shown the rate of presence of PAVM, CVM and treatment.
6. How about causes of death of two groups?

Discussion
7. Please discuss why the life expectancy of unscreened HHT patients was decreased, to show importance of screening.

Reviewer 5 Report

The authors presented a very interesting and valuable study on life expectancy for HHT patients. There are some concerns that the authors should consider to address. 

(1) in the abstract, the conclusion sentence should be elaborated by clarifying that those HHT patients who have been systematically screened and treated for HHT related organ involvement showed similar life exp. as compared to the control group.

(2) what is the life quality score for the HHT patients (systematically screened and treated) vs non-HHT patients? It will be interesting to know; at least it should be discussed if there is a data available in literature.

(3) What are the complications (commodities) for those patients in this cohorts? This should be included in table 1 for both groups. These variables should be adjusted in your analysis. 

(4) how many patients in the cohort have been treated in a conservative way or an aggressive way or not being treated? This information should be included in Table 1. Those patients who have been aggressively treated will be having similar life exp.? These subgroup of patients can be considered to be analyzed separately.

(5) Line 92-94, the known family HHT mutations should be exempted from this criteria since those clinically confirmed patients may harbor yet-to-be-determined novel mutations accounting for HHT. At least, this should be discussed as a limitation in the manuscript.